# Social network responses to victims of potentially traumatic events: A systematic review using qualitative evidence synthesis

**Marieke Saan**[1]*, **Floryt van Wesel**[1], **Sonja Leferink**[2], **Joop Hox**[1], **Hennie Boeije**[3], **Peter van der Velden**[4,5]

**1** Department of Methodology & Statistics, Faculty of Social and Behavioural Sciences, Utrecht University, Utrecht, The Netherlands, **2** Victim Support Netherlands, Utrecht, The Netherlands, **3** Nivel, Netherlands Institute for Health Services Research, Utrecht, The Netherlands, **4** CentERdata, Tilburg, The Netherlands, **5** TRANZO, Tilburg School of Social and Behavioral Sciences, Tilburg University, Tilburg, The Netherlands

* m.c.saan@uu.nl

**Data Availability Statement:** All relevant data are within the paper and its Supporting information files.

## Abstract

### Background

A substantial number of qualitative studies examined how adult victims of potentially traumatic events (PTEs) experienced support provided by family members, friends, colleagues, and other significant others in the informal network. Importantly, the large majority of qualitative studies focused on the perceived support of victims of specific events such as sexual offences, partner violence, homicide, accidents and disasters. Although it is likely that across specific PTEs there are similarities as well as differences in experienced support from the informal network, to date no systematic review synthesized the results of qualitative studies on support from the informal network following various types of PTEs. The aim of the present systematic review is to fill this gap in the scientific knowledge, which is also highly relevant for victim services, policymakers, and the informal network.

### Methods

A literature search of qualitative studies was conducted using the electronic databases of PubMed, Web of Science, CINAHL, Psych INFO, Scopus, Criminal Justice Abstracts and Picarta. The quality of the identified studies was assessed with the Consolidated Criteria for Reporting Qualitative research (COREQ) checklist, followed by analysis of the results of the identified studies using Qualitative Evidence Synthesis.

### Findings

Seventy-five papers were included in the synthesis, involving 2799 victims of PTEs such as accidents, disasters, homicide, intimate partner violence (IPV), and sexual offences. Saturation was only achieved for IPV. Overall, four major categories of perceived social support were identified, namely, support perceived as supportive, supportive but insufficient, unsupportive, and absent from informal support providers, which included friends, family, neighbors, (if applicable) offender's family, religious group members, work/school colleagues,

**Funding:** This study was made possible by a grant of the Victim Support Fund (Fonds Slachtofferhulp), The Hague, The Netherlands. The funders had no role in study design, data collection and analysis, decision to publish, or preparation of the manuscript.

**Competing interests:** The authors have declared that no competing interests exist.

fellow victims, the local community, and the social network in general. Across the PTE groups, there were similarities in experiencing positive forms of support (particularly *empathy* and *sharing experiences*) as well as negative forms of support (*abandonment*, *avoidance*, *lack of empathy*, and not experiencing support despite victim's request for help). There were also differences across PTE groups, in particular, victims of sexual and intimate partner violence mentioned a number of other supportive (*mobilizing support*, *no unsupportive responses*) and non-supportive (e.g., *justification* or *normalization* of violence and *minimizing* responses) responses.

## Conclusions

The review showed that different actors within the social informal network can play an important role in providing support after victims experience violence, homicide, accidents, and disasters. However, the review revealed that the large majority of qualitative studies were aimed at victims of IPV, and only for this type of PTE was saturation achieved. This indicates that, although this synthesis identified several similarities and differences, it is still too early to draw more definitive conclusions on similarities and differences in experienced social support after various PTEs and that future qualitative studies focusing on other PTEs are much needed.

## Introduction

Each year many adults are confronted with one or more potentially traumatic events such as traffic accidents, crime, violence, and disasters [1–3]. In the aftermath of such events, victims may experience a wide range of problems that are often inter-related. These include mental health problems and mental disorders, problems within relationships, financial and legal problems, disabilities due to sustained injuries, loss of trust, and struggles with perceived injustice [1, 4–9].

Although, according to Hobfoll's Conservation of Resources Theory [10–12], victims actively cope with these events and try to restore their lost resources, social support from the informal network such as family, friends, and colleagues may be indispensable for victims [13–16]. Two types of social support can be distinguished. Perceived social support is the subjective evaluation of the expected and actual quality of available support, while received social support is the subjective evaluation of the amount of provided support [17, 18]. Social support from the informal network may help victims—as long as the provided support meet the victims' needs—to effectively cope with problems and thereby contribute to recovery. Importantly, victims are more likely to interact and share their experiences with persons within their informal network than with formal support providers, such as their general practitioner, police, and mental health professionals [4, 19].

However, research has shown that some victims lack social support or are confronted with negative support from the informal network, suggesting that there is room for improvement [20–24]. A better fit between victims' needs for social support and the provided support is highly relevant because research has also shown that a perceived lack of support or negative support is associated with higher stress levels and PTSD symptom levels [17, 20, 25–27], while ongoing stress and PTSD symptom levels over the long term may erode social support levels [28, 29].

Given the differences between various types of PTEs and their consequences, it is likely that part of the problems victims face and their subsequent needs for social support from the

informal network are specifically related to the type of PTE and the victim's situation [10–12]. For instance, the need for temporary housing after a devasting flood will be absent after a traffic accident. Some needs, such as the need for a listening ear and social acknowledgement, may be more general and not specifically related to the type of PTE [30, 31]. In addition, the type of event may trigger specific responses from the informal network regardless of the needs of victims. For example, after violence-related events the informal network may express thoughts about the role and responsibility of the victims leading up to these events, while such thoughts will be absent after disasters that have natural causes, such as hurricanes.

Further insight into the differences and similarities in experienced social support across different types of PTEs is highly relevant for victim services, policymakers, and the informal network. This insight may help to improve the social support provided to victims by optimizing the match between needs and social support. The outcomes of both qualitative and quantitative PTE-related studies are of relevance in this perspective. However, PTE-related empirical quantitative studies on social support have predominantly focused on associations between support on the one hand and post-event mental health (especially PTSD symptomatology) on the other hand [20, 28]. These studies assessed the effects of different, general types of support, such as emotional, instrumental, informational, and esteem support. Little to no attention was paid to specific types of support following specific events, or the support received from specific support providers. When victims' needs and circumstances vary considerably, the same general forms of support may be less appropriate, and more understanding is needed of what is perceived by different types of victims as supportive and unsupportive of their informal network. Qualitative PTE-related research on social support after trauma did focus on perceived levels of support from different support providers and the different types of responses victims have received, but this research often focuses on specific groups of victims, such as victims of sexual offences [32], accidents [33], disasters [34], and intimate partner violence [35], and as a result, there remains little understanding of the experiences of different types of victims.

To the best of our knowledge, a systematic comparison of victim's perceptions of received social support across victims of accidents, disasters, and crime is nonexistent. In addition, systematic comparisons of different support providers within the informal network are absent. The aim of the present systematic review is to fill this gap in the scientific knowledge. By using the method of systematic review, the results of prior qualitative studies on social support are organized and summarized, allowing a comparison of social support experiences across several PTE groups. Synthesizing evidence stemming from qualitative research, also referred to as qualitative evidence synthesis, has become a more common practice in recent years [36, 37]. Compared to the findings of a single study alone, the synthesis of the findings of several relevant studies provides a more comprehensive perspective on a certain topic and identifies potentially conflicting finding [38]. This allows us to get a more comprehensive understanding of how different types of victims experience social support after a PTE and which informal network actors they consider important after experiencing such an event.

Considering the aforementioned, the research question of the current qualitative evidence synthesis is: *How do adult victims of violence, homicide, accidents and disasters experience responses from informal support providers in the aftermath of the event, and what are similarities and/or differences in experiences across various PTE groups according to qualitative studies*?

## Methods

### Literature search

In order to conduct the literature search, relevant keywords related to the four main components of the research question were identified: victim, type of potentially traumatic event,

**Table 1. Key words used for electronic data base search.**

| Components review question | Search string |
|---|---|
| Victim | ((victim* OR eyewitness OR survivor*) AND |
| Type of potentially traumatic event | (incident* OR assault* OR trauma* OR abus* OR violen* OR crim* OR rape OR stalk* OR loverboy OR groom* OR human trafficking OR sexual trafficking OR domestic violen* OR domestic abus* OR intimate partner violen* OR IPV OR honor related violen* OR honor killing* OR murder OR homicide OR manslaughter OR missing person OR theft OR burglar* OR robb* OR property crim* OR fraud* OR swindl* OR blackmail OR threat* OR kidnap* OR hostage* OR traffic OR road OR accident* OR crash* OR epidemic* OR fire OR storm OR flood OR disaster OR earthquake OR hurricane OR tornado OR medical malpractice OR medical mistake) AND |
| Social informal network | (volunt* OR citizen* OR nonprofessional* OR informal* OR amateur* OR hands-on expert OR spous* OR peer* OR family OR community OR social support* OR social network* OR unofficial) AND |
| Qualitative research | (qualitative OR mixed-method* OR unstructured interview* OR open interview* OR semi-structured interview* OR focus group* OR grounded theory OR grounded theories OR ethnograph* OR etnograf* OR ethnograf* OR phenomelogic* OR hermeneutic* OR life history* OR life stor* OR participant observation* OR open interview OR thematic analyses OR content analyses OR observational methods OR constant comparative method OR field notes OR field study OR audio recording)) |

social informal network, and qualitative research. We decided to use rather broad search terms in order to prevent that relevant qualitative studies using (slightly) different words would not be identified. Because of the changing criteria of PTSD in the DSM and ICD, we included the term trauma* besides several other event-related search terms. The search string is shown in Table 1.

We conducted a systematic search, without specific filters, in the following seven electronic databases: PubMed, Web of Science, CINAHL, Psych INFO, Scopus, Criminal Justice Abstracts, and Picarta. All searches were imported in EPPI Reviewer, specialized software for research synthesis [39], in order to organize and select the search records. The initial search was performed in October 2014 and was updated in November 2016 and February 2019.

## Inclusion criteria and study selection

For the purpose of this review, studies were included if they were written in English, were published in peer reviewed journals, and used qualitative data collection method(s) as well as qualitative data analysis method(s). Mixed-methods studies were also included when it was possible to extract findings from the qualitative research component [40]. Furthermore, this review was limited to peer-reviewed studies conducted after 1980 when PTSD was first codified in the DSM (DSM-III; [41]).

The focus of our review was on different types of potentially traumatic events such as crime, disasters, and accidents. Although our search was broad, we excluded the following studies:

1. Studies not performed among the general adult population, such as studies among rescue workers, soldiers, or veterans. Because of the nature of their work, as a group they are more often confronted with potentially traumatic events than the general population. In addition, there may be work-related (preventive) interventions that may change the demand for support from the (informal) network.

2. Studies with combined samples of victims and offenders or formal support providers.

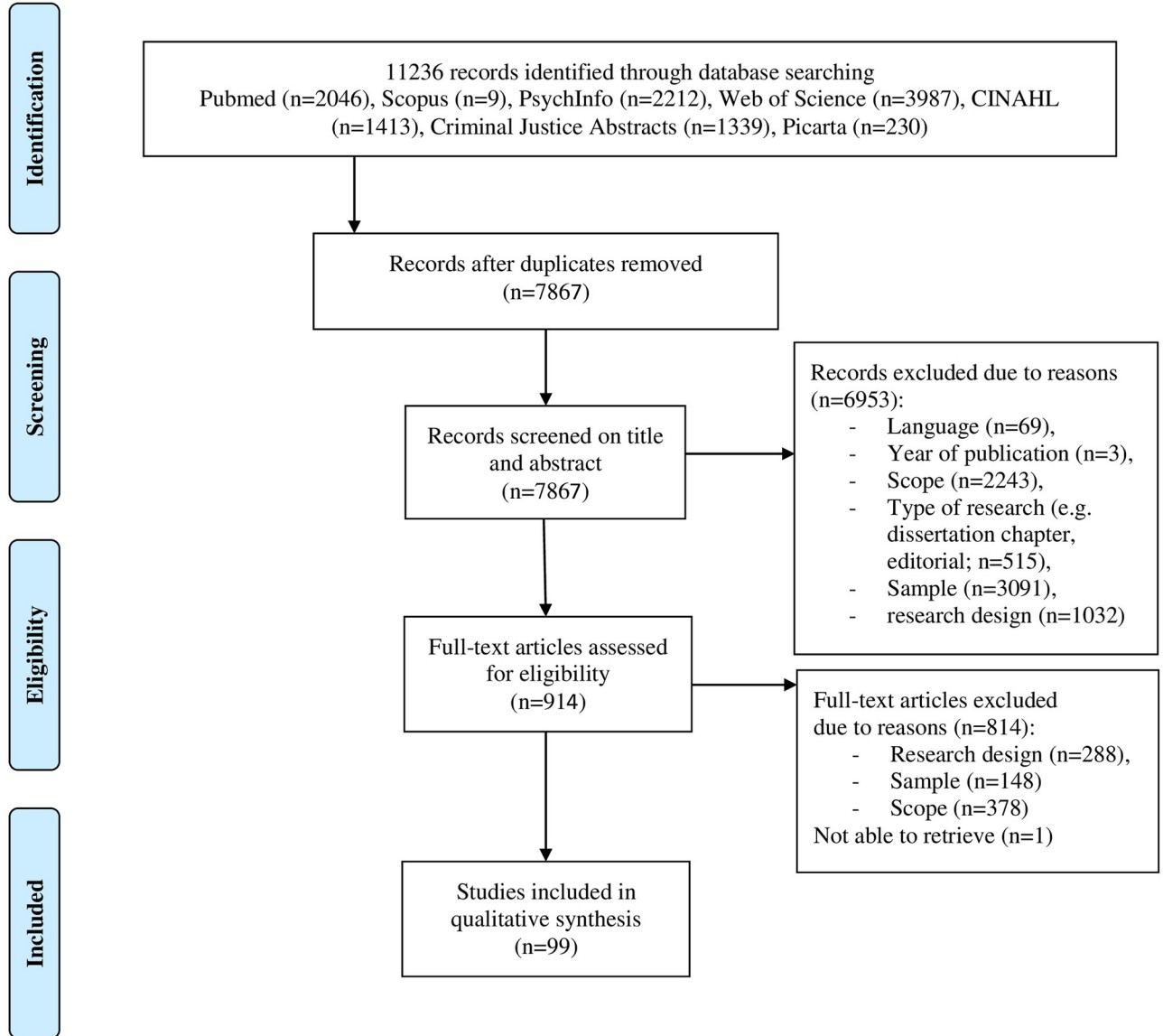

**Fig 1. PRISMA flow diagram of the literature search.**

3. Studies that focused on children.

4. Studies of which the main focus was not on victims' experiences of social support from their informal network (for example, studies on specific interventions or with a primary focus on alcohol/substance abuse). The results of the literature search are presented in Fig 1 below.

The search of the seven databases identified 11236 studies. After removal of duplicates, 7867 studies remained. After screening the titles, 3239 studies were excluded due to the language of the study (any language other than English), the year of publication (published before 1980), scope of the study (for example, studies on specific interventions), sample (such as

rescue workers) and type of publication research (for example, a conference paper or dissertation). The abstracts of the remaining 4628 studies were screened. A total of 3714 studies were excluded for reasons of sample, research design and scope of the study. Subsequently, the remaining 914 full texts were screened, and 814 studies were excluded for reasons of research design, type of research, sample or scope of the study. Title/abstract and full text screening was performed by MS. It was not possible to retrieve the full text of one paper, despite our attempts to contact the author. Finally, a total of 99 studies remained.

## Quality appraisal

Many authors have proposed the need to appraise the quality of qualitative research when conducting a qualitative evidence synthesis, mainly because the trustworthiness of the synthesis may be affected by the quality of the primary studies. However, there is no consensus about how quality should be assessed or how to deal with studies identified as being of 'poor' quality [42]. Therefore, we decided to critically appraise the included studies but not to exclude studies based on quality. MS assessed the reporting of the studies using the consolidated criteria for reporting qualitative research (COREQ; [43]) checklist. This reporting guideline has been deemed appropriate to assess the completeness of reporting and can be used to assess the methodological quality of the included studies and helps to interpret the findings [44]. In case of an interpretative uncertainty, consultation was sought within the research team.

## Data extraction and synthesis

Data analysis was conducted using NVivo 11 software for qualitative data analysis [45]. After the full texts of the included studies had been entered in NVivo 11, studies were arranged and coded according to the nature of events, resulting in various PTE groups. Furthermore, some basic study characteristics (author(s), year, country of origin, data collection method, time since event, and sample characteristics) were extracted. Then, full texts of all studies were searched for findings relevant to our review question. The focus was on findings in the results section, but findings that were part of the conclusion or discussion were also included. Inductive open coding techniques were used, and all relevant text fragments were assigned a particular theme (or code). Information about the nature of responses victims experienced after the PTE as well as the support providers were coded. When victims mentioned a response without referring to a specific support provider, it was coded as a response from the social network in general. If victims from a single study mentioned more than one type of support, each response was coded. This coding process was performed per group predominantly by MS, and it was discussed till consensus with HB and FvW was reached. The coding process within each group of PTEs was stopped when five subsequent studies were coded into pre-existing codes only and no additional codes would be derived from including more studies, i.e., saturation was reached.

After finalizing the process of open coding, the initial codes from one study within one group of PTEs were merged and refined, resulting in concepts. These concepts were then compared with the next randomly assigned study in the same group to identify similarities and differences. In each step, new concepts were created if concepts that already had been identified did not cover this concept. This process was repeated for all studies within a group of PTEs. Table 2 provides an example of the process of coding and collapsing a concept. In one study from the homicide group, the concept of *feeling pressured to get on with life* was coded. After comparing the codes of all studies within this group, this concept was identified in four studies (see Table 2). The process of merging and comparing initial codes into concepts was repeated

**Table 2. Example of coding and collapsing the concept of *feeling pressured to move on*.**

| Group | Illustrative quotes | Coded concepts | Final concept |
|---|---|---|---|
| Homicide [46, 47, 49, 54] | Some mothers felt subtle pressure to curtail or terminate their grieving before their internal process was completed. [49, quote author] | Feeling a time limit on grieving a deceased person [46, 49] | Feeling pressured to move on |
| | One mother succinctly stated, 'you know people tend to say, why don't you move on?' [49, quote victim] | Feeling pressured to get on with life [46, 47, 49, 54] | |
| Disasters [63, 65, 69] | "For six participants, it was because they felt that their significant others had at some point indicated that they ought to move on in life, either explicitly ("are you not well yet?") or implicitly ("it's so long ago now")." [63] | Feeling pressured to get on with life [63, 65, 69] | |
| Accidents [93] | One man said: People tell you that you only lost "things," and they say it like it was trash, "things!" They expect you to pull up your socks and get on with it. [93] | Feeling that grief over the loss of material things was not allowed [93] | |
| | Victims were told they should "Get on with it," "Pull yourself together," and "It's best not to talk about it." [93] | Feeling pressured to get on with life [93] | |
| Sexual offences [75, 76] | Another survivor described a friend who "would tell me to let it be in the past." [75, quote victim] | Feeling pressured to get on with life [75, 76] | |
| Intimate partner violence | *Not reported* | | |
| Various | *Not reported* | | |

for all groups. This led to an overview of all concepts related to the responses victims received for each group of PTEs.

The next step was to determine how the concepts from each victimization group were related across all groups. Because the IPV group had the largest body of literature, this group was the starting point for the analysis. The concepts found in this group were compared with the concepts in the next group to identify similarities and differences between groups. When necessary, concepts were collapsed or refined. This iterative process was repeated for all groups. Considering the example mentioned above, the concept of feeling pressured to get on with life was also identified in the disasters group, the accidents group and the sexual offences group (see Table 2). The process of merging and comparing concepts across victimization groups, led to an overview of all concepts including similarities and differences across the various PTEs.

After finalizing this process, MS and FvW reviewed all concepts and then revisited and refined all concepts until all data were explained and accounted for. Furthermore, all inconsistencies, contradictions, and definitions were intensively discussed by the whole research team. This resulted in a final refinement of the concepts. In case of our aforementioned example, the concept was merged with the concepts of *feeling a time limit on grieving over a deceased person* and *feeling that grief over loss of material things was not allowed* into the final concept of *feeling pressured to move on* (see Table 2).

## Results

The study findings are presented in two separate subsections. In the first subsection the literature search outcomes, study characteristics and COREQ assessment are described. The second subsection shows the synthesized support experiences of victims presented with illustrative quotations from the original studies.

### Search outcomes and study characteristics

As described above, the literature search identified 99 papers. These papers were categorized into different groups according to the type of PTE. We identified the following events (n = number of studies): accidents (n = 6), disasters (n = 19), homicide (n = 9), sexual offences (n = 17), various types of traumatic events within one study (n = 2), and intimate partner violence (IPV) (n = 46). For this final group, we reached saturation after analyzing 22 papers;

therefore, 24 papers were not included in the analysis. We did not reach saturation in the other groups. Consequently, from the original 99 papers, 75 papers were included in the synthesis.

Study characteristics per group—i.e., first author, year of publication, country of origin, data collection method, time since event, and sample of each study—are summarized in Table 3. Each study was given a unique reference number. The 75 papers represent 72 studies. Papers 77 and 84 and papers 79 and 80 share the same (sub)sample and one paper is part of one study published in two papers; one paper contains the introduction and methods sections, while the other paper contains the results and discussion sections [82, 83]. Table 3 shows that, although our search included studies published since 1980, the first identified study was published in 1993. The 72 unique studies are covering the experiences of 2799 victims of various PTEs including 254 family members, friends, or partners of victims. The studies originated from 26 countries, mostly from the United States (n = 42), Australia (n = 5), and Sweden (n = 5). The time between the event and the study ranges from 1 month to 30 years and is not always reported in the studies (n = 37). In the IPV group, it was or could not always be reported because some victims were still in the abusive relationships. Most studies used semi-structured interviews and focus groups to collect data. Other data collection methods that were used include surveys, field notes, expert interviews, observations, and archival records and other official data or documents.

It must be noted that, despite the use of various search terms related to the research questions, no qualitative studies on support following human trafficking, missing person, kidnapping, theft, robbery, burglary, medical mistakes, or fraud were found.

## COREQ assessment

The COREQ checklist was completed for 71 papers since 4 papers used a questionnaire as the main data collection method and not every item of the checklist applied. The completeness of reporting varied across studies, with studies reporting an average of 18 of the 32 items (range 11–32) from the COREQ checklist. The lowest rates of reporting were observed in domain 1 (research team and reflexivity), specifically in the subdomain relationships with participants. Establishing a relationship with the participants, participants knowledge of the interviewer and their characteristics regarding bias, assumptions and interest in research topic were reported in less than 30% of the studies. Greater transparency was apparent in domains 2 and 3. In domain 2 (study design), all studies reported the sample size and most reported the methodological orientation and theory, description of the sample, interview guide and audio/visual recording. The highest rates of reporting were observed in domain 3 (analysis and findings) in the subdomain reporting. In all studies there was consistency between the presented data and the findings and all used quotations to illustrate the findings (sometimes without reference due to the anonymity of participants). The full COREQ assessment is displayed in S1 Table.

## Experiences of support

Throughout the included studies, victims mentioned a variety of experiences with support from their social network in the aftermath of potentially traumatic events. Victims mentioned responses of others being **supportive**, **supportive but insufficient**, **unsupportive**, or **absent**. Victims often described a mix of responses, as a result of which the level of support varied. The various responses are explained in more detail in the following subsections. Table 4 provides an overview of the responses victims mentioned in relation to the support provider. The term 'support provider' is used for all people within the social network that victims mentioned, whether they responded positively and/or negatively. The responses are displayed in the rows and the different support providers that were mentioned in the context of this response are

**Table 3. Study characteristics included studies per group.**

| Nr. | First author (Year) | Country of Origin^ | Data collection* | Time between event and study | Participants |
|---|---|---|---|---|---|
| **Trauma group: homicide (n = 9)** | | | | | |
| [46] | Mastrocinque (2015) | USA | F.G. | M 11,16,3 years | 28 family and friends |
| [47] | Armour (2002) | USA | OE.I. | M 7,5 years | 14 family members |
| [48] | Englebrecht (2016) | USA | F.G. | M 11 years | 18 family members |
| [49] | Hannays-King (2015) | CAN | I. | range 0–8 years | 10 family members |
| [50] | Parapully (2002) | USA | Q. & SS.I. | Mdn 6 years | 16 family members |
| [51] | Sharpe (2008) | USA | SS.I. & FN. | M 13,8 years | 5 family members |
| [52] | Sharpe (2011) | USA | SS.I. | *homicides '96- '06* | 8 family members |
| [53] | Sharpe (2013) | USA | SS.I. | *not reported* | 12 family members |
| [54] | Baliko (2008) | USA | SS.I. | range 0–5 years | 10 family members or partners |
| **Trauma group: disasters (n = 19)** | | | | | |
| [34] | Binder (2014) | USA | SS.I, F.G., O., AR., MR., EI. | 1,5 years | 37 tsunami victims |
| [55] | Ekanayake (2013) | LKA | ID.I. & OSD. | 3–4 years | 38 tsunami victims |
| [56] | Ibañez (2004) | MEX USA | UI. | 3 months & 5 years | 10 sewer explosion & 17 hurricane |
| [57] | Ibañez (2003) | MEX USA | SS.I. | 3 months & 5 years | 9 sewer explosion & 16 hurricane |
| [58] | Doğulu (2016) | TUR | SS.I. | 2 years | 20 earthquake victims |
| [59] | Tirgari (2016) | IRN | SS.I. | 11 years | 12 family members earthquake |
| [60] | Tuason (2012) | USA | I. | 9 months | 9 hurricane victims |
| [61] | Rabelo (2016) | LBR | F.G. | *data Febr-April '15* | 17 victims of ebola virus disease |
| [62] | Doohan (2014) | SWE | I. | 1 month | 56 bus crash victims |
| [63] | Arnberg (2013) | SWE | I. & D.A. | 15 years | 22 ferry disaster victims |
| [64] | Forsberg (2011) | SWE | N.I. | 4 years | 14 train crash victims |
| [65] | Sample (2012) | USA | I., FN. & MR. | *happened in '95* | 20 bombing disaster |
| [66] | Woods (2014) | AUS | OE.S | 6–12 months | 433 cyclone victims |
| [67] | Becker (2015) | ZAF | SS.I. | 9 months | 7 bush fire victims |
| [68] | Jang (2009) | TWN | ID.I & F.G. | *happened '99* | 23 earthquake victims |
| [69] | Cho (2017) | KOR | SS.I. | 22–26 months | 54 family members of victims of ferry disaster |
| [70] | Brockie (2017) | AUS | ID.I. | few months and 2 yrs post floods '11, '13 | 10 flood victims |
| [71] | Cui (2017) | CHN | ID.I. | 6 months post event | 10 earthquake victims |
| [72] | Heid (2017) | USA | OE.I. | *happened '10* | 20 hurricane victims |
| **Trauma group: sexual offences (n = 17)** | | | | | |
| [32] | Ahrens (2007) | USA | SS.I. | *not reported* | 102 rape victims |
| [73] | Gutzmer (2016) | USA | SS.I. & Q. | *age in years 23–44, events since 18+* | 19 sexual coercion victims |
| [74] | Ahrens (2012) | USA | SS.I. | *not reported* | 103 sexual assault victims |
| [75] | Ahrens (2009) | USA | SS.I. | *not reported* | 103 sexual assault victims |
| [76] | Dos Reis (2016) | BRA | SS.I. | Table 1 *(not reported)* | 11 sexual assault victims |
| [77] | Duma part II (2007) | ZAF | ID.I. | within 6 months | 10 sexual assault victims |
| [78] | *Duma part I–this paper contains the introduction and methods section of the above paper.* | | | | |
| [79] | Ahrens (2006) | USA | SS.I. | M 16,10 | 8 rape victims |
| [80] | Onyango (2016) | COG | I. | *dc in '12, sexual violence since '96* | 55 sexual violence victims |
| [81] | Filipas (2001) | USA | OE.S. | average 10 years | 323 sexual assault victims |

(*Continued*)

**Table 3.** (Continued)

| Nr. | First author (Year) | Country of Origin^ | Data collection* | Time between event and study | Participants |
|---|---|---|---|---|---|
| [82] | Wadsworth (2018) | USA | SS.I. | *oct '14-april '15 interviews* | 22 sexual assault victims |
| [83] | Dworkin (2018) | USA | SS.I. | 21.3 months since assault | 26 sexual assault victims |
| [84] | Jackson (2017) | USA | I. | *Events in past year* | 18 sexual assault victims |
| [85] | Lorenz (2018) | USA | SS.I. | *not reported* | 45 sexual assault victims |
| [86] | Opsahl (2017) | USA | SS.I. | *not reported* | 3 sexual assault victims |
| [87] | Fileborn (2019) | AUS | Q. | *not reported* | 292 street harassment victims |
| [88] | Mahlstedt (1993) | USA | Q. | *not reported* | 103 victims of dating violence |
| **Trauma group: accidents (n = 6)** | | | | | |
| [33] | Moi (2014) | NOR | I. | 5–35 months postburn | 14 burn injury victims |
| [89] | Martin (2017) | AUS | SS.I. | *events at least 2 years previously* | 16 burn injury victims |
| [90] | Pashaei (2016) | IRN | SS.I. | max 2 years after | 18 victims of road traffic accidents |
| [91] | Tan (2008) | SGP | ID.I. | until 6 months after discharged hospital | 6 victims of motor vehicle accidents |
| [92] | Ravindran (2013) | IND | SS.I. | 6 weeks to 6 years | 22 family members of 12 burn injur |
| [93] | Stern (1996) | CAN, USA, DNK, SWE, AUS, KOR, CHN, NZL, FJI | I. | *data collection '87-'92* | 113 burn injury victims |
| **Trauma group: various (n = 2)** | | | | | |
| [94] | Anderson (2017) | USA | SS.I. | *not reported* | 16 victims of physical or sexual assault or accident |
| [95] | Nevhutalu (2014) | ZAF | F.G. | *not reported* | 75 victims of road accidents, rape, domestic abuse, and housebreaking |
| **Trauma group: intimate partner violence (n = 22)** | | | | | |
| [35] | Flinck (2005) | FIN | I. | *not reported* | 7 victims |
| [96] | Bui (2003) | USA | SS.I. | *some still in AR* | 34 victims |
| [97] | Lafferty (2013) | IRL | SS.I. | *all abuse had stopped* | 9 victims of domestic violence |
| [98] | Liendo (2011) | USA | SS.I. | *still in AR* | 26 victims |
| [99] | Loke (2012) | CHN | SS.I. | *all still in AR* | 9 victims |
| [100] | Ahmad (2013) | CAN | SS.I. | *all left AR* | 11 victims |
| [101] | Moe (2007) | USA | SS.I. | *living in shelter* | 19 victims of domestic violence |
| [102] | Morrison (2006) | USA | SS.I. | *> 1 year since the end of relationship* | 15 victims |
| [103] | Postmus (2014) | USA | SS.I. | *experienced IPV since start program* | 25 victims |
| [104] | Ridell (2009) | CAN | ID.I. & Q. | M 15,38 months since end of relation. | 9 victims (qualitative part) |
| [105] | Rose (2000) | USA | SS.I. | *15 still in AR* | 31 victims |
| [106] | Shen (2011) | TWN | SS.I. | *all participants had ended relation* | 10 victims |
| [107] | Ruijiraprasert (2009) | THA | I. | *6 still in AR* | 16 victims |
| [108] | Agoff (2007) | MEX | SS.I. | *majority still in AR* | 26 victims |
| [109] | Kyriakakis (2014) | USA | SS.I. & Q. | recent history of IPV (past 12 months) | 29 victims |
| [110] | Roush (2016) | USA | I. | Current or <3 years history with IPV | 12 victims |

(*Continued*)

**Table 3.** (Continued)

| Nr. | First author (Year) | Country of Origin^ | Data collection* | Time between event and study | Participants |
|---|---|---|---|---|---|
| [111] | Cox (2009) | USA | SS.I. | majority stalked in previous 5 years | 9 victims of stalking |
| [112] | Bornstein (2006) | USA | I. & F.G. | time since abuse 0.2–20 years (M 3,5) | 22 victims |
| [113] | Walters (2011) | USA | SS.I. | *no IPV in current relationship* | 4 victims |
| [114] | Bostock (2009) | ENG | I. | *not reported* | 12 victims of domestic violence |
| [115] | Crandall (2005) | USA | SS.I. & F.G. | *not reported* | 24 victims of domestic violence |
| [116] | Fiene (1995) | USA | UI. | *residing shelter* | 8 victims |

^Country of origin ISO abbreviation 3 code

*in order of occurrence; I. = interviews, F.G. = focus groups, OE.I. = open-ended interviews, ID.I. = in-depth interviews, Q. = Questionnaire, SS.I. = semi-structured interviews, FN. = field notes, UI. = unstructured interviews, OSD. = official sources data, D.A. = diagnostic assessment, SS.Q. = semi-structured questionnaire, O. = observations, AR. = archival records, MR. = media records, EI. = expert interviews, NI = narrative interviews, OE.S. = open-ended survey

shown in the columns. Fellow victims refers to individuals sharing the same or a similar potentially traumatic event. When victims mentioned a response without referring to a specific support provider, the response is displayed as a response from the social network in general. The type of potentially traumatic event after which victims received the response from the support provider is displayed in the sub columns. The number in a cell of the table corresponds to the number of studies in which the response was mentioned. The response was counted if at least one victim in the study mentioned this response. For example, if a disaster victim reported experiencing both empathy and practical help, this study was included twice in the number of studies shown under the disasters sub column. Supplementary material contains an expanded version of this table in which each cell contains the superscript numbers that refer to the reference numbers of the original studies (see S2 Table).

## Supportive responses

In almost all studies, victims mentioned experiencing supportive responses (or helpful, positive, and healing) from their informal network in the aftermath of potentially traumatic events. Supportive responses were given by several support providers, as displayed in Table 4. Family and friends were mentioned most often, yet other members of the social informal network were also mentioned, such as neighbors, members and leaders of a religious community, work/school colleagues, and fellow victims. The various supportive responses that victims mentioned varied from receiving *advice*, receiving *information*, and *companionship*, to the *absence of unsupportive responses*. Responses were characterized as supportive because they made victims feel they were cared for, understood, and accepted [32, 33, 47, 58]. They also made victims feel less isolated and relieved, as though "a great burden had been lifted from their shoulders" [46, 48, 59, 75, 96, 107]. Furthermore, responses were seen as supportive when they helped to validate and normalize victim's feelings [32, 75, 105]. When they received helpful responses, victims perceived that their feelings about the incident were acknowledged. Feelings such as "I don't deserve this, I deserve better" were reinforced [105].

Consequently, supportive responses helped victims to cope with the consequences after a potentially traumatic event. Emotional and practical support helped victims to better deal with their emotions and practical issues such as financial problems. Victims explicitly mentioned

**Table 4. Overview of responses victims experienced in relation to support provider.**

| | Family | | | | | Friends | | | | | Work/school colleagues | | | | |
|---|---|---|---|---|---|---|---|---|---|---|---|---|---|---|---|
| | Accidents | Disaster | Homicide | IPV | Sexual offences | Accidents | Disaster | Homicide | IPV | Sexual offences | Accidents | Disaster | Homicide | IPV | Sexual offences |
| **Type of supportive response** | | | | | | | | | | | | | | | |
| Practical help | 2* | 9 | | 9 | 3 | 2 | 5 | 1 | 8 | 4 | 1 | 2 | | | 1 |
| Information | | 1 | | 1 | 1 | | 2 | | 1 | 2 | | | | | |
| Emotional help | 2 | 8 | | 8 | 4 | | 7 | 1 | 6 | 6 | | 2 | | | |
| Advice | | | 1 | 2 | 1 | | | 1 | | 1 | | | | | |
| Companionship | | 4 | 2 | | | | 1 | | | | | | | | |
| Empathy | | 2 | 1 | | 3 | | 1 | | 1 | 1 | | | | | |
| Attempt to intervene | | | | 4 | | | | | 1 | | | | | | |
| Listening ear | | 5 | | | 1 | | 3 | 1 | 1 | 2 | | | | | |
| Mobilizing support | | | | 5 | 3 | | | | 5 | 2 | | | | 1 | |
| No unsupportive responses | | | | 1 | 2 | | | | | 2 | | | | | |
| Respect autonomy | | | | | | | | 1 | 1 | | | | | | |
| Safety | | | | | 1 | | | 1 | | | | | | | |
| Seeking Justice | | 1 | | | 2 | | 1 | | | | | 1 | | | |
| Sharing experiences | | 1 | 2 | | 2 | | 1 | | | 5 | | | | | |
| Solidarity | | 1 | 3 | | | | | | | | | | | | |
| Unconditional support | 1 | | 2 | 1 | | 1 | | | 1 | | | | | | |
| Validation | | | | 2 | 4 | | | | | 4 | | | | | 1 |
| **Type of insufficient response** | | | | | | | | | | | | | | | |
| Insufficient practical help | 1 | | | 1 | 1 | | | | 1 | 1 | | | | | |
| Insufficient emotional help | 1 | 2 | 1 | 2 | 1 | 1 | | | | 2 | | | | 1 | |
| **Type of unsupportive response** | | | | | | | | | | | | | | | |
| Abandonment | | 3 | 3 | 1 | 3 | 2 | 1 | 2 | 1 | 1 | | 2 | | | |
| Avoidance | | 2 | 2 | 1 | 1 | 1 | | 2 | 2 | | | | | | |
| Blaming | 1 | | | 6 | 10 | | | | 3 | 8 | | | 1 | | 1 |
| Complicating responses | | | | 1 | 2 | | | | 1 | 2 | | 1 | 1 | | |
| Egocentric responses | | 2 | 4 | | 7 | | | | | 7 | | | | | |
| Feeling pressured to move on | | 1 | 2 | | 2 | | 1 | 1 | | 1 | | | | | |
| Justification | | | | 7 | | | | | 3 | | | | | | |
| Minimizing | | | | 5 | 5 | | | | 4 | 5 | | | | 1 | |
| Lack of empathy | 2 | 1 | | 2 | 2 | 2 | 1 | | 2 | 1 | | 1 | 1 | | 1 |
| No attempt to intervene | | | | 5 | | | | | 3 | | | | | | |
| Not respecting autonomy | 1 | | | | 6 | 1 | | | 1 | 2 | | 1 | | | 2 |
| Treat different | | | 1 | | 3 | | | | | 2 | | | | | |
| **Absence of response** | | | | | | | | | | | | | | | |
| No support | | 1 | 3 | 4 | 2 | | | 3 | 2 | 2 | | | | | |

*(Continued)*

**Table 4.** (Continued)

| | Neighbors | | | | | Fellow victims | | | | | Religious group members | | | | |
|---|---|---|---|---|---|---|---|---|---|---|---|---|---|---|---|
| | Accidents | Disaster | Homicide | IPV | Sexual offences | Accidents | Disaster | Homicide | IPV | Sexual offences | Accidents | Disaster | Homicide | IPV | Sexual offences |
| **Type of supportive response** | | | | | | | | | | | | | | | |
| Practical help | 2 | 6 | | | | | | | 1 | | | 3 | | | |
| Information | | 1 | | | | | | | 2 | | | | | | 1 |
| Emotional help | | 4 | | | | | 3 | | 1 | | | 2 | | 1 | |
| Advice | | | | | 1 | | | | | | | | | | |
| Companionship | | | | | | | 1 | | | | | | | | 1 |
| Empathy | | | | 1 | | | | 4 | | | | | | | |
| Attempt to intervene | | | | | | | | | | | | | | | |
| Listening ear | | 1 | | | | | 1 | 1 | | | | 1 | | | 1 |
| Mobilizing support | | | | 1 | | | | | | | | | | | |
| No unsupportive responses | | | | | | | | | | | | | | | |
| Respect autonomy | | | | | | | | | | | | | | | |
| Safety | | | | | | | | 2 | 1 | | | | | | |
| Seeking Justice | | | | | | | | | | | | | | | |
| Sharing experiences | | | | | | 1 | 8 | 3 | 2 | 1 | | 1 | | | |
| Solidarity | | | | | | | 1 | | 1 | | | | | | |
| Unconditional support | | | | | | | | | | | | | | | |
| Validation | | | | | | | | | | | | | | | |
| **Type of insufficient response** | | | | | | | | | | | | | | | |
| Insufficient practical help | | | | | | | 2 | | | | | | | | |
| Insufficient emotional help | | | | | | | | 1 | | | | | | | |
| **Type of unsupportive response** | | | | | | | | | | | | | | | |
| Abandonment | | | | | | | | | | | | 1 | | | 1 |
| Avoidance | | | | | | | | | | | | | | | |
| Blaming | | | | | | | | | | | | | | 1 | 2 |
| Complicating responses | | | | 1 | | | | | | | | | | | 1 |
| Egocentric responses | | | | | | | | | | | | | | | |
| Feeling pressured to move on | | | | | | | | | | | | | 1 | | |
| Justification | | | | | | | | | | | | | | 3 | |
| Minimizing | | | | | | | | | | | | | | | |
| Lack of empathy | | | | 1 | | | | | | | | | | | 1 |
| No attempt to intervene | | | | 1 | | | | | | | | | | | |
| Not respecting autonomy | | | | | | | | | | | | | | | |
| Treat different | | | | | | | | | | | | | | | |
| **Absence of response** | | | | | | | | | | | | | | | |
| No support | | | | | | | | | | | | | | | 2 |

(*Continued*)

**Table 4.** (Continued)

| | Community | | | | | Offender's family | | | | | Social network in general | | | | |
|---|---|---|---|---|---|---|---|---|---|---|---|---|---|---|---|
| | Accidents | Disaster | Homicide | IPV | Sexual offences | Accidents | Disaster | Homicide | IPV | Sexual offences | Accidents | Disaster | Homicide | IPV | Sexual offences |
| **Type of supportive response** | | | | | | | | | | | | | | | |
| Practical help | 1 | 7 | | 1 | | | | | 1 | | 4 | 12 | 1 | 11 | 5 |
| Information | | | | 2 | | | | | | | 1 | 2 | 1 | 4 | 4 |
| Emotional help | | 5 | | 1 | 2 | | | | | | 2 | 13 | 1 | 9 | 8 |
| Advice | | 1 | | 2 | 1 | | | | | | | 1 | 1 | 5 | 2 |
| Companionship | | 1 | | | 1 | | | | | | 1 | 5 | 2 | | 1 |
| Empathy | | | 1 | 1 | | | | | | | 1 | 4 | 4 | 2 | 7 |
| Attempt to intervene | | | | 1 | | | | | 2 | | | | | 6 | |
| Listening ear | | 3 | | 2 | | | | | | | | 9 | 2 | 3 | 5 |
| Mobilizing support | | | | | | | | | | | | | | 7 | 5 |
| No unsupportive responses | | 1 | | | | | | | | | | | | 1 | 4 |
| Respect autonomy | | | | 1 | | | | | | | | | 1 | 1 | |
| Safety | | | | | | | | | | | | | 3 | 1 | 1 |
| Seeking Justice | | | | | 1 | | | | | | | 1 | | | 3 |
| Sharing experiences | | | | | | | | | | | 1 | 8 | 4 | 2 | 5 |
| Solidarity | | 9 | | | | | | | | | | 9 | 3 | 1 | 1 |
| Unconditional support | | | | | | | | | | | 2 | | 2 | 1 | |
| Validation | | | | 1 | | | | | | | | | | 3 | 9 |
| **Type of insufficient response** | | | | | | | | | | | | | | | |
| Insufficient practical help | | | | | | | | | | | 1 | 2 | 1 | 1 | 1 |
| Insufficient emotional help | | 1 | | | | | | | | | 3 | 5 | 1 | 3 | 3 |
| **Type of unsupportive response** | | | | | | | | | | | | | | | |
| Abandonment | 2 | | | | 1 | | | | | | 2 | 3 | 4 | 1 | 6 |
| Avoidance | | | | 1 | | | | | 1 | 1 | 1 | 2 | 3 | 4 | 2 |
| Blaming | | | | 2 | 1 | | | | 3 | 1 | 2 | | 2 | 12 | 13 |
| Complicating responses | | 2 | | 2 | | | | | 4 | | | 2 | 1 | 6 | 5 |
| Egocentric responses | | | | | | | | | 1 | | | 2 | 4 | 1 | 10 |
| Feeling pressured to move on | | | | | | | | | | | 1 | 3 | 4 | | 2 |
| Justification | | | | 1 | 1 | | | | 3 | | | | | 9 | 1 |
| Minimizing | | | | 2 | 1 | | | | | | | | | 8 | 9 |
| Lack of empathy | 1 | 1 | 1 | | 1 | | | | | | 4 | 2 | 3 | 2 | 4 |
| No attempt to intervene | | | | | | | | | | | | | | 6 | 1 |
| Not respecting autonomy | | | | 1 | | | | | | | 1 | 1 | | 2 | 8 |
| Treat different | | | 1 | | | | | | | | | | 1 | | 4 |
| **Absence of response** | | | | | | | | | | | | | | | |
| No support | 1 | | 1 | | | | | | 1 | | 2 | 1 | 3 | 5 | 3 |

*The number in a cell corresponds to the number of studies in which the response was mentioned.

that support was important for the recovery process [68, 82, 84, 90, 91, 100]. Support helped them to get through, promoted resilience and was a strong contributor to survivorship [50, 58, 59, 62, 64, 67, 92]. Furthermore, feeling supported strengthened existing relationships [33, 47, 48, 50, 53, 54, 66, 74, 82]. Several victims mentioned stronger family ties. Supportive responses from others also made it easier to seek and/or accept further (formal) help, such as from the police [105, 107, 114].

**Similarities and differences across PTE groups.**   Several supportive responses were mentioned in all PTE groups. *Practical help* and *emotional help* in general were perceived as supportive by victims from all groups. Receiving practical help (such as goods and money) and psychological, emotional, or moral help were found to be supportive. In several PTE groups, three types of practical help were mentioned as being particularly supportive: childcare [50, 75, 92, 100, 102, 104, 109, 114], financial support [33, 55, 57, 67, 70, 72, 92, 101, 107, 109, 114], and shelter [34, 57, 65, 72, 92, 93, 96, 100, 102, 104, 107, 109, 114]. Another response that was found to be supportive across PTE groups was *sharing experiences*, e.g., sharing experiences with individuals who experienced the same of similar type of PTE. Through sharing feelings about their experiences, victims felt understood. This feeling of understanding stems from the idea that "they have been there". It also provided a way to put together "pieces of the puzzle". Victims expressed the feeling that no one can provide support as well as those who have experienced such an event themselves. Sharing experiences was sometimes facilitated by organized and non-organized support groups. Another recurring supportive response mentioned across the PTE groups was experiencing a *listening ear*. Victims spoke of the importance of having the opportunity to tell and retell their stories and support providers taking the time to listen to those stories.

However, in addition to the supportive responses mentioned across different PTE groups, Table 4 also displays responses mentioned in only one or two groups. For example, *(attempt to) intervene* was only mentioned in the IPV group. Victims experienced support when others made efforts to help the victims get out of the abusive relationship or confronted the abuser with their behavior. Another PTE-specific response was observed in the sexual offences group. Victims mentioned several responses that may not seem supportive such as blaming, doubting, or controlling responses, but were nevertheless perceived as supportive when victims believed that the intention of the supporters was good [75, 83, 85]. We coded these responses as *well-intended responses*. PTE-specific results related to particular support providers also emerged. Heid et al. [72] described the important role that neighbors can play after someone has experienced a disaster. Neighbors provided a greater variety of support and assistance than family and friends. In the IPV group, family members of the offender were specifically mentioned as providing supportive responses.

## Supportive but insufficient responses

In all PTE groups, some victims reported receiving support but experienced this support as insufficient [49, 54, 58, 61, 67, 72, 92, 93, 98, 101, 105, 106, 116]. Family, friends, work/school colleagues, fellow victims, and the community were mentioned in this context. Victims were not completely satisfied with the amount or quality of support they received. Sometimes the help of others was perceived as inadequate or indifferent, and victims had the feeling that others were not showing enough interest or concern when providing support. Support was also experienced as being insufficient when it was limited to one source of support. Because support was sometimes perceived as insufficient, victims experienced little comfort; they mentioned it affected their ability to cope with the event and it made them feel alone, among other things.

## Similarities and differences across PTE groups

*Insufficient emotional help* was mentioned in all PTE groups. Victims reported that while they had experienced some (often practical) support, they perceived it as minimally emotionally supportive or lacked emotional support completely. In the accidents, disasters, and homicide group, it also emerged that victims did not experience enough long-term support. This gap between victim's expectations versus received support developed over time. Though support was provided immediately after the incident, once the crisis passed victims experienced a withdrawal of support, even though they still needed it [49, 54, 67, 72, 93]. In the disasters and homicide group, it was mentioned that when informal support providers were involved in the event themselves, they were less able to be supportive. In the disasters and sexual offences group, two specific experiences emerged. In the disasters group, victims mentioned that sharing experiences with fellow victims was perceived as less supportive when there was too much diversity in the experiences of the group members. In the sexual offences group, it was described that the network's responses were not perceived as sufficient due to the severity of the event. Another response mentioned in the context of insufficient support across several groups (specifically the accidents, disasters, and IPV group) was *insufficient practical help*. Victims experienced a lack of practical help in general terms, and in the accidents and disasters group it was also more specifically mentioned in context of insufficient health care.

## Unsupportive responses

Victims also mentioned receiving unsupportive (or hurtful, unhelpful and negative) responses from their informal network, with family and friends most often mentioned. Only fellow victims were not mentioned in this context. In addition, several victims had to deal with unsupportive responses from various actors in their social network at the same time. As a result, victims had little resources left. Table 4 shows all unsupportive responses, varying from receiving *blaming* responses to *lack of empathy* or *feeling pressured to move on*. Responses were found to be unsupportive when they made victims feel disappointed, hurt, betrayed, and sorrowful [32, 33, 35, 46, 49, 63, 75, 84–86, 89, 99, 113]. Responses were also characterized as being unhelpful when they made victims feel isolated, socially estranged, and alone [46, 47, 49, 60, 61, 69, 70, 105, 113, 114, 116]. Sometimes victims mentioned even feeling revictimized due to unsupportive responses [84, 86, 107, 113]. They were recalling the event and felt violated again [84, 89]. In some studies, victims mentioned inappropriate support attempts including insensitive responses, insults, or inappropriate advice [33, 54, 79, 89, 92, 93, 102, 106].

Consequently, due to unsupportive responses, the victims' own negative feelings about the incident such as self-blame, guilt, embarrassment, or shame were reinforced [49, 75, 76, 79, 84, 86, 92, 101, 102, 107, 114]. Unsupportive responses also had the effect of deteriorating relationships [32, 34, 46–49, 53, 54, 58, 60, 74, 82, 84, 85, 89]. Victims mentioned family and community relationships being strained and several relationships being lost after unhelpful responses. In addition, negative responses also had the effect of inhibiting further disclosure or seeking help [49, 79, 84–86, 89, 102, 107, 113, 114]. Unhelpful responses during initial disclosure resulted in victims ceasing to disclose altogether, silencing the victims. Thus, victims indicated that unsupportive responses had various far-reaching consequences.

**Similarities and differences across PTE groups.** Several unsupportive responses were experienced by victims of multiple groups. For example, experiencing a *lack of empathy* was noted as being unsupportive in all PTE groups. These responses were experienced as lacking sensitivity, sympathy, compassion or understanding. Furthermore, *abandonment* was experienced by victims from all groups. Victims experienced rejection and felt abandoned by people from their social network. Family and friends distanced themselves and suddenly disappeared.

Another recurring unsupportive response experienced across several PTE groups (except for the disasters group) was *blaming*. This response was coded when victims experienced blaming responses (like holding victims responsible for the traumatic events) or judging responses (like making a mental judgement about the event). When experiencing these responses, victims felt they shared culpability for the traumatic event. For example, they were blamed for being in a certain situation or for (not) showing certain behavior.

Besides unsupportive responses that were experienced across PTE groups, Table 4 also shows unsupportive responses mentioned in only one or two groups. For example, *justification* was only mentioned in the IPV group and the sexual offences group, referring to responses that justified or normalized the violence. Examples of such responses include suggestions that woman (in general) deserve to be beaten, or that according to others the incident was a normal reaction to something the victim did [106]. Another response that was especially mentioned in the IPV group was *complicating responses*, i.e., responses or actions which further burdened victims. These responses included support providers who jeopardized a victim's safety by telling their location to the perpetrator or acted violently themselves after victims disclosed their experiences.

### Absence of responses

Besides the supportive and unsupportive responses described above, some of the victims in all PTE groups reported not receiving any response at all. Potential support providers did not respond to victims' attempts to seek help or actively refused to help; therefore an attempt to seek help did not result in any kind of support [32, 46, 47, 49, 70, 75, 79, 89, 92, 95, 98, 101, 104, 105, 113]. The absence of responses was mentioned in the context of the community, family, friends, offender's family, and members of religious organizations. The fact that victims did not receive any response even though they had expressed the need for help to others made them feel unsupported. Among other things, this resulted in them experiencing more stress and led to impairment of social support systems.

**Similarities and differences across PTE groups.** The absence of responses was experienced by victims from all groups. In several groups (but not the IPV group or disasters group), victims reported experiencing a lack of support from support providers because they did not know what to say or do [46, 49, 54, 89]. In two groups, in the IPV group and sexual offences group, victims mentioned support providers actively refusing help. In addition, in the IPV group one specific consequence of the absence of support was mentioned. Several studies in this group reported that as a result of informal support providers not providing any support the victims felt isolated and therefore were not able to leave the abusive relationship [104, 105, 116].

### Discussion

The research question of the present systematic review was how do adult victims of violence, homicide, accidents, and disasters experience responses from informal support providers in the aftermath of the event, and what are similarities and/or differences in experiences across various PTE groups according to qualitative studies?

The literature search identified 99 qualitative papers that met our selection criteria. Seventy-five qualitative papers published until February 2019 were identified and analyzed. Studies focused on victims' experiences of various PTEs, namely accidents, homicide, disasters, sexual offences, and IPV. Despite our broad search terms, no qualitative studies were identified that focused on the experiences of social support from the informal network of victims of human trafficking, missing persons, kidnapping, theft, burglary, robbery, medical mistakes, or

fraud. In addition, the large majority of the identified studies (n = 46) focused on victims of IPV and saturation was reached for this group only. This means that much of what we know is based on the experiences of these victims and that we do not yet have a complete picture of the experiences of other victims. Therefore, we can only draw preliminary conclusions about similarities and differences between victim experiences, and further research is needed. The synthesis did reveal specific supportive, unsupportive, insufficient, or lack of responses, as well as a several similarities and differences among the different PTE groups. Some of these similarities and differences were not unexpected and seem to be explained by the type of event the victims experienced, while others were more salient. We also found that the included qualitative studies did not always distinguish between different support providers, as well as existing social support questionnaires, and paid little attention to the interaction between informal support providers and victims. However, the results of the synthesis suggest that it is important to consider different support providers because specific support providers seem to play a particularly positive or negative role after a victim experiences a PTE. First, the different experiences of victims and several important similarities and differences will be explained, after which the aforementioned gaps in existing research will be addressed.

## Victims' support experiences after experiencing a PTE

The qualitative evidence synthesis revealed that victims perceived the support of the informal network after experiencing a PTE as supportive, insufficient, unsupportive, or absent and mentioned several specific (un)supportive responses that characterized these experiences. Several potential support providers in the informal network were identified: friends, family, neighbors, offender's family (if applicable), fellow victims, religious group members, work/school colleagues and the local community. Victims mentioned experiencing different consequences of supportive and non-supportive responses. Supportive responses helped them cope with the consequences of a PTE, strengthened relationships, and helped them to seek and accept further (professional) help. These findings are in line with previous research demonstrating the importance of informal support after a PTE in helping victims to recover or protect their resources [13–15, 117]. However, victims who did not receive support or experienced unsupportive responses mentioned feeling more stressed, hurt, and alone, and said that their negative feelings, such as guilt and shame about the incident, were reinforced. They also mentioned the effect of deteriorating relationships and reluctance to seek further help from (formal) support providers. These findings are consistent with previous research showing that unsupportive interactions with one's social network induce or reinforce stress [20–23].

**Similarities in victims' experiences with informal social support.** Across the different PTE groups, similarities were found in what responses and support providers victims experienced as supportive or unsupportive. Practical, emotional, and informational help were perceived as supportive, and in this context various actors in the informal network were identified, with family and friends being particularly mentioned. Unsupportive responses that emerged across all PTE groups included insufficient emotional help, lack of empathy, feeling abandonment, avoidance, as well as no support. Several other unsupportive responses emerged in multiple PTE groups, such as insufficient long-term support (in the accidents, disasters & homicide group), blaming (all but the disasters group) and feeling pressured to move on (all but the IPV group). Unsupportive responses were also experienced from various actors in the informal network, with fellow victims being the only support provider not mentioned in the context of unsupportive responses.

The importance of emotional support for victims after a PTE has been demonstrated in multiple quantitative and qualitative studies, as well as in our synthesis. Experiencing a

listening ear and being able to share experiences with others who had experienced the same or similar event were two specific types of emotional support that victims in all groups found supportive. This finding is consistent with previous research showing that it can be important for victims to process their experiences into stories so that they can cognitively better grasp and give meaning to their experiences. The informal network can play an important role in this by offering an unconditional listening ear. Moreover, by sharing experiences with people who have gone through a similar experience, victims can perceive that certain emotions or questions related to the event are normal [118, 119]. The synthesis showed that sharing experiences was sometimes facilitated by (non-) organized support groups, but also occurred during social interactions in everyday life. However, empirical studies on fellow-victim support for people who have gone through a traumatic experience are scarce [120] and future research will need to examine how effective fellow-victim support is and when it can be most supportive.

Furthermore, the synthesis revealed multiple similarities between unsupportive responses experienced by victims from different PTE groups. An interesting finding was that not only did relatives or friends of homicide victims experience they were being pressured to move on or were told that their grief was not valid, but this was also reported by victims of sexual offences, disasters, and accidents as well. Victims in these groups also mentioned the feeling of being pressured to move on and victims (in the accidents and disasters group) felt their grief over material items was not acknowledged. Furthermore, existing research on victims of sexual offences and IPV often reports victims receiving blaming responses [19, 121]. In our synthesis, blaming responses were indeed mentioned by these victims, but they were also experienced by victims in the accidents and homicide group. Victims were accused of living in "the wrong part of town" or were asked, "what did you do?" Perhaps this is because some people believe that victims, not just victims of sexual offences, must have made mistakes that played a causal role in their PTE [122].

**Differences in victims' experiences with informal social support.** Several differences emerged between the PTE groups in what victims considered supportive or unsupportive; some of these differences were not unexpected and seem to be explained by the type of PTE. For example, in the IPV group, the offender's family played both a supportive and unsupportive role. Unsurprisingly, in other groups this provider was not mentioned, since there was not always a (known) perpetrator involved. Among disasters victims, neighbors appeared to be especially important; although they were often affected by the disaster themselves, they were crucial in providing support, followed by family members. Perhaps victims found their neighbors helpful specifically because they had experienced the same event, which is consistent with the finding above that sharing experiences with fellow victims was perceived as being very helpful.

A specific finding in the sexual offences group was that victims mentioned certain responses that appeared unsupportive (e.g., blaming, despairing, or controlling) but were sometimes interpreted as supportive. Victims indicated that they perceived these responses as supportive because they were given by family members and friends and viewed them as signs of caring or acting in their best interests. This suggests that the perception of a response may depend on the context in which it is given, by whom it is given, and on the victim's interpretation of it [17].

Finally, several unsupportive responses emerged that were experienced by both victims of sexual offences and IPV (but not by victims from the other groups), such as support providers justifying or normalizing the violence and minimizing or questioning the event. Victims in these groups also mentioned that support providers sometimes actively refused to provide help. Perhaps potential support providers lack knowledge about how to respond supportively after experiencing such a PTE or perhaps victims of sexual offences and IPV are more upset (react more emotionally) causing the support provider to also have a strong emotional reaction that they have difficulty controlling [123].

## Gaps in current research and implications for future research

**Selective attention within social support research.** Despite our extensive search, we found no qualitative studies regarding social support for several other potentially traumatic events such as robbery or fraud. As previously mentioned, saturation was only reached in the IPV group, so the results of this review must be interpreted with caution. There appears to be selective attention within qualitative social support research with considerable attention to victims of IPV and less attention to victims of other PTEs.

Prevalence rates of lifetime PTEs do not appear to explain these inequalities. Kessler et al. [2017; [124]] reported the lifetime prevalence rates from the WHO Mental Health Survey (n = 68894) with rates for physical abuse by partner of 4,5%, while rates for natural disaster, automobile accident, and robbery ranged from 7,45 to 14,5%. According to Kelly [2011; [125]] the scholarly attention to IPV increased significantly as funding for research and practice increased. In addition, the continued development of feminism contributed to an increase in research. According to feminist analysis, IPV aims to maintain male dominance in the social environment [126]. However, there is some criticism in the field of IPV that most research focuses on female victims in heterosexual relationships, while violence against men and within same-sex relationships also occurs. Our review contributes to this finding because none of the included studies included male victims and only two studies included same-sex relationships. Thus, within the field of IPV, future research should focus on these less studied topics. More importantly, there is a need for further attention to several other PTE groups including victims of human trafficking, robbery, and accidents.

**Interaction between support provider and victims.** Another important gap that emerged in this study is that of the 75 studies included in our synthesis, only one study assessed the interaction between informal support providers and victims. This study [85] concluded that victims and support providers do not always have the same interpretations of social responses. For example, one victim experienced both an empathetic and revenge response from a friend, while that friend did not remember the revenge response and only remembered a supportive response. This finding is in line with previous quantitative research suggesting that victims and support providers may not always perceive disclosure in the same way [21]. Although support providers tried to offer support, victims did not seem to feel supported or were hurt by the response. Due to the lack of the support providers' perspectives, it is unknown to what extent victims' perceptions match those of providers. Understanding the similarities and differences between perceived received and perceived provided support is important because improving this interaction will ultimately improve the support victims receive in the aftermath of a PTE. Future research on social support should therefore be complemented with the experiences of informal support providers to examine interaction between victims and support providers. Furthermore, the results of the study of Lorenz et al. [2018] showed that disclosure itself, may have a negative impact on the support provider causing feelings such as distress and anger. It is possible that support providers are reacting negatively to victims because they themselves have been affected [24, 127, 128]. Research that directly addresses the impact of traumatic events on the informal social network is scarce [129].

**Challenging the measures of social support.** In addition, some of our review findings may challenge the way social support is measured in quantitative research. Although questionnaires commonly used to measure perceived social support often do not take the support providers into account (cf. [130]), our findings suggest that it is relevant to distinguish between support providers. It seems that victims found some responses to be supportive or unsupportive of specific support providers (and possibly seeking certain types of help from specific support providers). For example, neighbors and the community were particularly involved in

providing practical help and friends were involved in providing information. Insufficient emotional support as well as egocentric responses were often experienced by family members. Moreover, the results suggest that the type of support provider seemed to influence the victim's perception of the support. For example, certain responses that appeared to be unsupportive were still experienced as supportive because the providers were family members or friends.

The synthesis showed a variety of support experiences, with victims often mentioning a mix of supportive and unsupportive responses. However, several instruments measuring social support are measuring only one aspect of negative support (perceived criticism and feeling let down), do not measure negative support (Crisis Support Scale [131], [132]; Social Support Rating Scale [133]; Social Support Inventory [134]), or do not include victims' experiences of insufficient or no support (Social Reactions Questionnaire [135]; Unsupportive Social Interactions Inventory [134], [136]). To get a comprehensive overview, it may be important to use multiple instruments that measure different forms of support. Furthermore, it appears that several responses victims mentioned are not being questioned specifically (such as justification responses, responses without empathy and complicating responses). Thus, victims' experiences may not be fully covered by existing social support questionnaires due to current classifications and the failure to take the support provider into account. However, as mentioned above, findings of this systematic review must be interpreted with caution because saturation was only achieved in the IPV group.

## Limitations

The results of this review of qualitative studies need to be considered within the context of its limitations. First, it is important to be aware of the fact that only papers in the English language were included and that most of the included studies were carried out in Western countries. This raises questions about the transferability of our findings to other cultural contexts. Second, as mentioned earlier, saturation was only reached for the studies on IPV, so attritional responses may be unidentified. Third, we only searched in electronic databases to identify relevant studies. This may have left some research unidentified, for example the study of Edwards, Dardis & Gidycz [137]. However, the results of this mixed-method study on the experiences of 44 victims of dating violence show the same supportive and unsupportive responses as identified in the synthesis. In particular, a listening ear and emotional support were experienced as supportive, while a lack of understanding and minimizing responses were experienced as unsupportive. Fourth, the identified studies focused on social support as perceived by victims. It is unclear to what extent perceived and provided support matched and/or why providers offered the support they provided. Fifth, this review has focused solely on adult victims and not on children. In order to identify what child victims experience as supportive and unsupportive after experiencing a traumatic event another review is required. Last, currently there is a world-wide pandemic of COVID-19. Our search was carried out before the outbreak, and it is therefore not part of this review.

## Conclusions

The results of this qualitative evidence synthesis demonstrated the great importance of different actors within the social informal network in providing support to victims of violence, homicide, accidents, and disasters. The synthesis showed specific forms of practical, informational, and emotional support that helped victims cope after PTEs, strengthened relationships, and facilitated the search for additional (professional) help. However, several forms of support were perceived as unsupportive or absent. These negative forms of support reinforced feelings of distress, strained relationships, and sometimes even silenced victims. Overall, the study revealed important similarities and differences between different PTE groups and support

providers, suggesting that it is important to consider different types of PTE in future research and that a distinction between informal support providers may also provide important insights. This may help to better align victims' needs and social support and may be of great importance for victim services and targeted interventions. In addition, this comprehensive systematic review of qualitative research on social support revealed several gaps in the existing literature, the most important of which is that the current knowledge from qualitative studies is based primarily on the social support experiences of victims of IPV. Future qualitative studies should investigate whether the patterns of experienced support found in this review can be confirmed.

## Supporting information

**S1 Table. COREQ assessment per study.**
(DOCX)

**S2 Table. Victims' experiences with references to included studies.**
(DOCX)

**S1 Checklist. PRISMA checklist.**
(PDF)

## Author Contributions

**Conceptualization:** Marieke Saan, Floryt van Wesel, Sonja Leferink, Joop Hox, Hennie Boeije, Peter van der Velden.

**Formal analysis:** Marieke Saan, Floryt van Wesel, Hennie Boeije.

**Funding acquisition:** Sonja Leferink, Hennie Boeije.

**Methodology:** Marieke Saan, Floryt van Wesel, Hennie Boeije.

**Project administration:** Marieke Saan.

**Supervision:** Floryt van Wesel, Hennie Boeije, Peter van der Velden.

**Writing – original draft:** Marieke Saan, Floryt van Wesel, Peter van der Velden.

**Writing – review & editing:** Marieke Saan, Floryt van Wesel, Sonja Leferink, Joop Hox, Hennie Boeije, Peter van der Velden.

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
