## [Decision Letter · Decision Letter 0]

30 Dec 2021

PONE-D-20-37652

Social network responses to victims of potentially traumatic events: A systematic review using Qualitative Evidence Synthesis

PLOS ONE

Dear Dr. Saan,

Thank you for submitting your manuscript to PLOS ONE. After careful consideration, we feel that it has merit but does not fully meet PLOS ONE’s publication criteria as it currently stands. Therefore, we invite you to submit a revised version of the manuscript that addresses the points raised during the review process.

The reviewers of your new submission feel that you have responded well to comments raised on your previous submission, but that there are a number of outstanding concerns that need to be addressed in order for your manuscript to meet PLOS ONE's publication criteria. Please ensure that you address each of the reviewers' outstanding concerns carefully when preparing your revisions.

We look forward to receiving your revised manuscript.

Kind regards,

Jamie Males

Staff Editor

PLOS ONE

Journal Requirements:

Reviewers' comments:

Reviewer's Responses to Questions

**Comments to the Author**

1. Is the manuscript technically sound, and do the data support the conclusions?

Reviewer #1: Yes

Reviewer #2: Partly

2. Has the statistical analysis been performed appropriately and rigorously? 

Reviewer #1: Yes

Reviewer #2: N/A

3. Have the authors made all data underlying the findings in their manuscript fully available?

Reviewer #1: Yes

Reviewer #2: Yes

4. Is the manuscript presented in an intelligible fashion and written in standard English?

Reviewer #1: Yes

Reviewer #2: Yes

5. Review Comments to the Author

Reviewer #1: I consider the authors have accomplished the reviewer´s comments and they have provided an improved version.

Just the first figure they have attached is not very clean (please attached a figure with better resolution).

Reviewer #2: Thank you for the opportunity to review this manuscript. This is a systematic review of qualitative studies, aiming to answer to the following research question: How do adult victims of violence, homicides, accidents and disasters experience responses of informal support providers in the aftermath of the event, and what are similarities and/or differences in experiences across various victimization groups according to qualitative studies?

Among the strengths of this study is an original and worthwhile aim (to synthetize qualitative studies on social support after trauma, with a focus on differences sources of social support) and inclusion of a large number of studies. I sincerely applaud the authors’ interest in contributing with something new in this field, and I agree that we need new development both within theoretical and methodological approaches to social support. At the time I read this manuscript, it has received feedback from two previous reviewers, and from what I can judge, this feedback has been taken seriously and has probably increased the quality of the manuscript.

I still have some major concerns:

1) The introduction does not provide a clear rationale for why it is necessary to study different groups of victims together. On what theoretical or empirical grounds do you expect differences in experience of information support providers? The introduction somewhat unfocused and includes various references suggesting that “it’s complicated” rather than a clear rationale for why your study has to be done.

2) The results are difficult to discern – there are many details and few superordinate take-home messages.

3) The discussion should be more connected to the theory and empirical studies mentioned in the introduction.

More detailed feedback:

Abstract:

- The background lacks an explicit and substantial objective – what is the research question of this paper? “Organizing and summarizing studies on” is a little passive. What does this study aim to teach us something about?

- I would like the findings part to be more specific and superordinate.

- I did not understand this sentence: “Various responses were experienced across victimization groups while others were more trauma-specific.”

- This sentence does not provide substantial information: “The synthesis provides several directions for future research and implications for victim assistance policies.” (which directions and implications do you mean?)

Introduction:

- The authors have chosen to base their study on Hobfoll’s conservation resource theory. Does this theory provide any input that can shed light on the research question? Please outline.

- The authors refers to several snippets of the social support literature. But I find the overall argumentation for the research question weak – do you expect that different types of support is more prevalent after some types of traumatic events – for example – people are more emotionally supportive after terrorism than after sexual abuse? Or that people offer more instrumental support after natural disasters than after sexual abuse? Or that people receive more unhelpful responses after family violence than after accidents? Please outline this link.

Methods:

- I find that the authors have described their methods in a thorough and accurate way.

Results:

- The four paragraphs based on the four topics are nicely written, but I would like some more substantial overall extractions. There are many details here, and the answer to the research question is not clearly presented.

- The tables are somewhat intimidating, at least for this reader. I would consider place table 2 in supplemental materials, and I would try to simplify table 4. It is very extensive and it is difficult to read. It looks more like an evidence gap map (https://eppi.ioe.ac.uk/cms/Default.aspx?tabid=3790 ) than a table with substantial results. So I look at it and I wonder: What can we learn from this? Perhaps this version of this table could be put in the supplemental materials as well?

Discussion:

- I am unsure what your main findings were. After you have reminded us about your research question, please formulate 3-5 main findings that you will discuss.

- I would like to see a discussion linking your findings to the theory and empirical studies you mention in the introduction, and integrate your findings it better in the literature. For example, I find it very interesting that you found that peers was the only support provider not mentioned as unhelpful. How would you interpret this in light of the literature showing that psychological debriefing after trauma are potentially harmful? Should we arrange for people to meet and share their experiences, or should we not?

- Please also write explicitly what your contribution to the literature was. I think that several of your findings could be results from a quantitative study as well.

6. PLOS authors have the option to publish the peer review history of their article (what does this mean?). If published, this will include your full peer review and any attached files.

Reviewer #1: No

Reviewer #2: **Yes: **Marianne Skogbrott Birkeland

---

## [Author Response · Author response to Decision Letter 0]

18 Jul 2022

We would like to thank the reviewers for taking the time to review and comment on the article. Attached to the submission is a letter explaining, point by point, how the reviewers' comments were addressed (Response to reviewers).

---

## [Decision Letter · Decision Letter 1]

19 Aug 2022

PONE-D-20-37652R1Social network responses to victims of potentially traumatic events: A systematic review using Qualitative Evidence SynthesisPLOS ONE

Dear Dr. Saan,

Thank you for submitting your manuscript to PLOS ONE. After careful consideration, we feel that it has merit but does not fully meet PLOS ONE’s publication criteria as it currently stands. Therefore, we invite you to submit a revised version of the manuscript that addresses the points raised during the review process.

I addition to the final minor comments provided by Reviewer 2 below, please address the following points in your response:

In the Introduction, lines 132 0 136 (“Qualitative PTE-related research on social support…”), please include some references to the studies you are referring to here.

In the introduction, paragraph 5 (lines 120 – 136), the authors refer to some PTE-relevant quantitative and qualitative studies that have examined social supports, but only state what they’ve focused on, rather than what they’ve found. Please provide a brief description of the main, collective findings from these studies and highlight what gaps remain.

In the ‘study selection’ section, lines 277 – 286, please include how many people were involved in the title/abstract screening and the full-text screening, it would also be useful to include the initials of the authors who undertook the screening at each stage. If possible, please also include what level of inter-rater agreement was reached for each stage, and how disputes in decisions were resolved.

In the Discussion, lines 642 – 644, it is unnecessary to restate the results in terms of number of papers found (e.g., “The synthesis identified 17 specific types of supportive responses, 12 non-supportive responses, …”). Please remove this. There are a few other sections within the Discussion where the number of papers are stated, and could be removed.

I would also caution the authors to be wary that they are not simply restating the Results within the Discussion, but instead are discussing what the results mean in the context of what literature has been previously published. A final review of the Discussion by the authors might reveal section where the word count could be cut down by removing the restating of results.

The ‘Challenging the measures of social support’ section in the Discussion reads more as an ‘implications’ section, rather than highlighting a gap in research – so perhaps make it its won implication section. The authors use ‘Furthermore’ a lot in this section, it feels quite repetitive and impedes the flow of the discussion. Please consider removing the ‘furthermores’.   

We look forward to receiving your revised manuscript.

Kind regards,

Michelle Torok, Ph.D.

Academic Editor

PLOS ONE

Journal Requirements:

Reviewers' comments:

Reviewer's Responses to Questions

**Comments to the Author**

1. If the authors have adequately addressed your comments raised in a previous round of review and you feel that this manuscript is now acceptable for publication, you may indicate that here to bypass the “Comments to the Author” section, enter your conflict of interest statement in the “Confidential to Editor” section, and submit your "Accept" recommendation.

Reviewer #2: All comments have been addressed

2. Is the manuscript technically sound, and do the data support the conclusions?

Reviewer #2: Yes

3. Has the statistical analysis been performed appropriately and rigorously? 

Reviewer #2: N/A

4. Have the authors made all data underlying the findings in their manuscript fully available?

Reviewer #2: Yes

5. Is the manuscript presented in an intelligible fashion and written in standard English?

Reviewer #2: Yes

6. Review Comments to the Author

Reviewer #2: Thank you for revising the paper in line with my suggestions. I find that the manuscript is much improved, in particular Table 4 which is now much more readable and informative. One potential remaining issue may be that literature search was updated the last time in February 2019, which is 3.5 years ago now. I will let it be up to the journal editor to decide whether there is a need to update it again.

Otherwise I only have minor issues that I spotted when I read through the paper now.

1. Abstract: The first sentence in the conclusion repeats what is already in the methods and I do not understand the second part. I would omit this sentence.

2. Table 3: Should it be “victims of ebola virus disease” rather than just “ebola virus disease”?

3. Table 3: It is not clear what “Duma part I – introduction a nd method section” mean.

4. Table 4: Please indicate what the numbers refer to in the title or note of the table (all tables should be self-explanatory).

7. PLOS authors have the option to publish the peer review history of their article (what does this mean?). If published, this will include your full peer review and any attached files.

Reviewer #2: No

---

## [Author Response · Author response to Decision Letter 1]

30 Sep 2022

Below we have described in detail how we responded to each comment. We start with the comments of the academic editor and then with the comments of reviewer 2. 

ACADEMIC EDITOR

1.In the Introduction, lines 132 0 136 (“Qualitative PTE-related research on social support…”), please include some references to the studies you are referring to here.

Response

1.Thanks to the editor’s comment, we included a reference for each PTE group mentioned in the sentence: 

Qualitative PTE-related research on social support after trauma did focus on perceived levels of support from different support providers and the different types of responses victims have received, but this research often focuses on specific groups of victims, such as victims of sexual offences [32], accidents [33], disasters [34], and intimate partner violence [35], and as a result, there remains little understanding of the experiences of different types of victims.

2.In the introduction, paragraph 5 (lines 120 – 136), the authors refer to some PTE-relevant quantitative and qualitative studies that have examined social supports, but only state what they’ve focused on, rather than what they’ve found. Please provide a brief description of the main, collective findings from these studies and highlight what gaps remain.

Response

2.We thank the editor for the suggestion. Earlier in the introduction we briefly describe the main findings of current research on social support: 

However, research has shown that some victims lack social support or are confronted with negative support from the informal network, suggesting that there is room for improvement [20-24]. A better fit between victims’ needs for social support and the provided support is highly relevant because research has also shown that a perceived lack of support or negative support is associated with higher stress levels and PTSD symptom levels [17,20,25-27], while ongoing stress and PTSD symptom levels over the long term may erode social support levels [28,29].

Later in the introduction, we explain the current gaps in quantitative and qualitative research and the importance of this study: 

These studies assessed the effects of different, general types of support, such as emotional, instrumental, informational, and esteem support. Little to no attention was paid to specific types of support following specific events, or the support received from specific support providers. When victims’ needs and circumstances vary considerably, the same general forms of support may be less appropriate, and more understanding is needed of what is perceived by different types of victims as supportive and unsupportive of their informal network. Qualitative PTE-related research on social support after trauma did focus on perceived levels of support from different support providers and the different types of responses victims have received, but this research often focuses on specific groups of victims, such as victims of sexual offences, accidents, disasters, and intimate partner violence, and as a result, there remains little understanding of the experiences of different types of victims.

We hope that with this we have sufficiently answered the editor's point.

3.In the ‘study selection’ section, lines 277 – 286, please include how many people were involved in the title/abstract screening and the full-text screening, it would also be useful to include the initials of the authors who undertook the screening at each stage. If possible, please also include what level of inter-rater agreement was reached for each stage, and how disputes in decisions were resolved.

Response

3.We thank the editor for the suggestion. We added the following sentence in study selection section: 

Subsequently, the remaining 914 full texts were screened, and 814 studies were excluded for reasons of research design, type of research, sample or scope of the study. Title/abstract and full text screening was performed by MS. 

Because the screening of titles, abstracts and fulltext was performed by MS there was no inter-rater agreement. In coding and synthesizing the articles, there was coordination with others on the research team. This is explained in the data extraction and synthesis section: This coding process was performed per group predominantly by MS, and it was discussed till consensus with HB and FvW was reached. & After finalizing this process, MS and FvW reviewed all concepts and then revisited and refined all concepts until all data were explained and accounted for. Furthermore, all inconsistencies, contradictions, and definitions were intensively discussed by the whole research team.

4.In the Discussion, lines 642 – 644, it is unnecessary to restate the results in terms of number of papers found (e.g., “The synthesis identified 17 specific types of supportive responses, 12 non-supportive responses, …”). Please remove this. There are a few other sections within the Discussion where the number of papers are stated, and could be removed.

Response

4. We thank the editor for the comments and removed the number of papers and changed the sentence:

The qualitative evidence synthesis revealed that victims perceived the support of the informal network after experiencing a PTE as supportive, insufficient, unsupportive, or absent and mentioned several specific (un)supportive responses that characterized these experiences

We also checked the rest of the discussion and adjusted it where necessary. 

5.I would also caution the authors to be wary that they are not simply restating the Results within the Discussion, but instead are discussing what the results mean in the context of what literature has been previously published. A final review of the Discussion by the authors might reveal section where the word count could be cut down by removing the restating of results.

Response

5. Thanks to the editor’s comments, we have reviewed the discussion and tried to further limit the repetition of results. Since this involves multiple changes, we refer the editor to the new discussion. 

6.The ‘Challenging the measures of social support’ section in the Discussion reads more as an ‘implications’ section, rather than highlighting a gap in research – so perhaps make it its won implication section. The authors use ‘Furthermore’ a lot in this section, it feels quite repetitive and impedes the flow of the discussion. Please consider removing the ‘furthermores’. 

Response

6. We thank the editor for the suggestion. We think the section does fit nicely in its current form because the gaps in current research are linked (both implicit and explicit) to implications for further research (see also the sections selective attention within social support research and interaction between support provider and victims). We added implications for future research to the subtitle of the section to make this more explicit. We also removed the use of furthermore where possible.

REVIEWER 2

1.Abstract: The first sentence in the conclusion repeats what is already in the methods and I do not understand the second part. I would omit this sentence.

Response

1.We thank the reviewer for the comment. We removed the sentence. 

2.Table 3: Should it be “victims of ebola virus disease” rather than just “ebola virus disease”?

Response

2.We thank the reviewer for the suggestion. We changed it to victims of ebola virus disease. 

3.Table 3: It is not clear what “Duma part I – introduction a nd method section” mean.

Response

3.We thank the reviewer for the suggestion. We explained it in the text (… one paper is part of one study published in two papers; one paper contains the introduction and methods sections, while the other paper contains the results and discussion sections [82,83].) but to clarify it, we have also explained it in more detail in the table: Duma part I – this paper contains the introduction and methods section of the above paper. 

4.Table 4: Please indicate what the numbers refer to in the title or note of the table (all tables should be self-explanatory).

Response

4.Thanks to the reviewer’s comments, we added the following note to the table: 

The number in a cell corresponds to the number of studies in which the response was mentioned

---

## [Editor Report · Decision Letter 2]

10 Oct 2022

Social network responses to victims of potentially traumatic events: A systematic review using Qualitative Evidence Synthesis

PONE-D-20-37652R2

Dear Dr. Saan,

We’re pleased to inform you that your manuscript has been judged scientifically suitable for publication and will be formally accepted for publication once it meets all outstanding technical requirements.

Kind regards,

Michelle Torok, Ph.D.

Academic Editor

PLOS ONE
---

## [Editor Report · Acceptance letter]

14 Oct 2022

PONE-D-20-37652R2 

Social network responses to victims of potentially traumatic events: A systematic review using Qualitative Evidence Synthesis 

Dear Dr. Saan:

I'm pleased to inform you that your manuscript has been deemed suitable for publication in PLOS ONE. Congratulations! Your manuscript is now with our production department. 

Kind regards, 

on behalf of

Dr Michelle Torok 

Academic Editor

PLOS ONE